# Whole-Blood Transcriptional Profiles Enable Early Prediction of the Presence of Coronary Atherosclerosis and High-Risk Plaque Features at Coronary CT Angiography

**DOI:** 10.3390/biomedicines10061309

**Published:** 2022-06-02

**Authors:** Daniele Andreini, Eleonora Melotti, Chiara Vavassori, Mattia Chiesa, Luca Piacentini, Edoardo Conte, Saima Mushtaq, Martina Manzoni, Eleonora Cipriani, Paolo M. Ravagnani, Antonio L. Bartorelli, Gualtiero I. Colombo

**Affiliations:** 1Centro Cardiologico Monzino IRCCS, 20138 Milan, Italy; eleonora.melotti@gmail.com (E.M.); chiara.vavassori@cardiologicomonzino.it (C.V.); mattia.chiesa@cardiologicomonzino.it (M.C.); luca.piacentini@cardiologicomonzino.it (L.P.); edoardo.conte@cardiologicomonzino.it (E.C.); saima.mushtaq@cardiologicomonzino.it (S.M.); martina.manzoni@cardiologicomonzino.it (M.M.); eleonora.cipriani@cardiologicomonzino.it (E.C.); paolo.ravagnani@cardiologicomonzino.it (P.M.R.); antonio.bartorelli@cardiologicomonzino.it (A.L.B.); 2Department of Biomedical and Clinical Science “Luigi Sacco”, University of Milan, 20121 Milan, Italy; 3Department of Clinical Sciences and Community Health, University of Milan, 20121 Milan, Italy; 4Department of Electronics, Information and Biomedical Engineering, Politecnico di Milano, 20133 Milan, Italy; 5Department of Biomedical Sciences for Health, University of Milan, 20121 Milan, Italy

**Keywords:** RNA sequencing analysis, circulating transcriptome, coronary CT, advanced plaque analysis

## Abstract

Existing tools to estimate cardiovascular (CV) risk have sub-optimal predictive capacities. In this setting, non-invasive imaging techniques and omics biomarkers could improve risk-prediction models for CV events. This study aimed to identify gene expression patterns in whole blood that could differentiate patients with severe coronary atherosclerosis from subjects with a complete absence of detectable coronary artery disease and to assess associations of gene expression patterns with plaque features in coronary CT angiography (CCTA). Patients undergoing CCTA for suspected coronary artery disease (CAD) were enrolled. Coronary stenosis was quantified and CCTA plaque features were assessed. The whole-blood transcriptome was analyzed with RNA sequencing. We detected highly significant differences in the circulating transcriptome between patients with high-degree coronary stenosis (≥70%) in the CCTA and subjects with an absence of coronary plaque. Notably, regression analysis revealed expression signatures associated with the Leaman score, the segment involved score, the segment stenosis score, and plaque volume with density <150 HU at CCTA. This pilot study shows that patients with significant coronary stenosis are characterized by whole-blood transcriptome profiles that may discriminate them from patients without CAD. Furthermore, our results suggest that whole-blood transcriptional profiles may predict plaque characteristics.

## 1. Introduction

Atherosclerotic cardiovascular (CV) disease still represents a major cause of mortality, despite a decline in incidence rates in several countries in Europe. Several risk scores have been proposed for the assessment of individual 10-year CV risk based on traditional risk factors (sex, age, smoking, total and LDL cholesterol, etc.) [1]. Nevertheless, existing tools to estimate CV risk have sub-optimal predictive capacities and the accurate identification of “at-risk” individuals remains a major challenge [2]. Coronary lesions that are not hemodynamically significant may be responsible for acute coronary syndromes (ACS). Thus, performing a well-planned clinical, biochemical, and genetic evaluation may be important to provide a complete CV risk assessment.

Noninvasive imaging techniques could improve risk-prediction models for CV events. In this setting, coronary CT angiography (CCTA) plays an important role in the identification of individuals with suspected coronary artery disease (CAD) and shows excellent diagnostic accuracy compared to invasive coronary angiography [3,4]. CCTA enables the early quantification of coronary atherosclerosis, before the onset of the clinically overt disease [5], but also the detection of adverse plaque characteristics associated with markers of histological vulnerability [6] and with prognosis in several studies [5,7].

In addition, investigation of molecular biomarkers (e.g., transcript biomarkers, proteins, or metabolites), which are present and altered in the biological fluids and at the level of the injured tissue [8], may help to reclassify patients according to various levels of risk, starting from a sub-clinical level of atherosclerosis up to high-risk subjects [9]. Blood is an ideal surrogate tissue for CAD studies [10,11]: tests on peripheral blood could be of paramount importance for CV disease diagnosis and prognosis. Due to the dynamic nature of the transcriptome, gene expression profiling has turned out to be an important research tool for elucidating disease pathophysiology and finding specific biomarkers.

Recently a few authors have used the RNA sequencing (RNA-Seq) technique to explore the whole-blood transcriptome in search of biomarkers of coronary atherosclerosis and/or related conditions. Preliminary studies have identified gene expression profiles that would distinguish women with acute myocardial infarction (MI) with nonobstructive CAD from those with MI-CAD or controls [12], and subjects with coronary stenosis ≥50% from those with stenosis <50% [13]. Two larger studies documented transcriptional expression signatures that discriminated a low degree of stenosis (≤20%) from an intermediate degree (>20% but <70% in any vessel) in patients presenting for elective coronary catheterization [14], and subjects with early MI from those with subclinical coronary atherosclerosis (detected as a high degree of coronary artery calcification (CAC)) and from controls without a history of MI or high CAC [15]. In all these studies, the presence or absence of coronary atherosclerosis had been documented by invasive coronary angiography.

Through the identification of biomarkers and non-invasive imaging analysis, it should be possible to monitor the development of atherosclerotic lesions, identify the “vulnerable patient”, and potentially prevent fatal outcomes.

This study makes use of CCTA and RNA-Seq to (1) identify whole-blood gene expression patterns that could differentiate patients with severe coronary atherosclerotic pathology from subjects with complete absence of CAD, (2) detect possible pathways and immune cell populations associated with the pathological phenotype through functional inference analysis, and (3) assess the association of gene expression patterns with CCTA-derived plaque features. It should be emphasized that this is a pilot study and proof-of-concept research that is intended to evaluate the experimental validity of an innovative hypothesis in a small number of subjects.

## 2. Materials and Methods

### 2.1. Study Design

The present work made use of the EPIFANIA prospective observational study underway at our Institute, which enrolled a large cohort of consecutive patients undergoing CCTA for suspected CAD, in order to identify molecular and imaging biomarkers of coronary atherosclerosis progression and occurrence of CV events. Inclusion criteria were: age between 18 and 80 years and negative history of previous coronary events (angina, silent ischemia, acute myocardial infarction) or revascularization (coronary angioplasty or bypass graft). Exclusion criteria were: age <18 and >80 years, pregnancy, history of previous CV or coronary revascularization, indication for immediate revascularization, arrhythmias, congestive heart failure, non-ischemic cardiomyopathies, pacemakers, defibrillators, severe or disabling extra-cardiac disease or infectious disease, immunosuppressive therapy or chemotherapy in the previous year, major surgery in the previous six months, blood perfusions in the previous two months, and failure to obtain the informed consent.

Patient enrollment and signing of informed consent, as well as peripheral blood sampling from an antecubital vein, were undertaken upon admission at the Department of Cardiovascular Imaging before CCTA was performed. Medical history and clinical information were collected. Routine hematochemical tests were performed along with the CCTA examination.

For this exploratory pilot study, we adopted a strategy that would allow us to search for highly relevant differences and remove possible confounding factors. Therefore, we selected two groups of individuals at the extremes of the distribution of coronary artery stenosis extent (i.e., patients with severe stenosis versus subjects who showed no signs of coronary atherosclerosis), matched for sex, age, and CV risk factors, from those enrolled between October 2016 and November 2018.

The study protocol conformed to the principle of the Declaration of Helsinki and was approved by the Ethics Committee of the IRCCS Istituto Europeo di Oncologia and Centro Cardiologico Monzino. All recruited patients signed the written informed consent form and participants agreed to share their de-identified information.

### 2.2. Coronary Computed Tomography Angiography

CCTA was performed using a novel scanner (Revolution CT, GE Healthcare, Milwaukee, WI, slice configuration 256 × 0.625 mm, gantry rotation time 280 msec, prospective ECG triggering) for qualitative and quantitative assessment of the atherosclerotic burden and detection of high-risk plaque features. Coronary plaques were defined as structures of at least 1 mm^2^ area within and/or adjacent to the artery lumen, clearly distinguishable from the vessel lumen. High-risk plaque features, such as the arterial remodeling index (RI), spotty calcification (SC), and napkin ring sign (NRS), were evaluated. The RI is defined as the ratio between the lesion plaque area and a reference lumen area. The NRS is a thin, rim-like enhancement (no more than 130 HU) distributed along the outer contour of the vessel and surrounding a fibro-lipidic plaque. SC is any discrete calcification ≤3 mm in length and occupying ≤90° arc when viewed on a short axis [16]. Plaque density was assessed using HU, and low attenuation plaque (LAP) was defined as the presence of any voxel < 30 HU. Total plaque volume (TPV) was evaluated and reported in mm^3^ [16,17]. Low attenuation plaque volume (<30 HU) and fibrotic plaque volume (30–150 HU), summarized as the noncalcific plaque volume, were defined as the amount of plaque with <150 HU [16]. In all patients, the segment involvement score (SIS) and segment stenosis score (SSS) were calculated. The SIS is the total number of coronary artery segments with plaques (ranging from 0 to 16) and reflects the extent of the plaques. The SSS is the sum of the extent grades based on the Coronary Artery Disease Reporting and Data System (CAD-RADS) (grade 0: no visible stenosis; grade 1: <25% stenosis; grade 2: 25–49% stenosis; grade 3: 50–69% stenosis; grade 4: 70–99% stenosis; and grade 5: totally occluded) of all segments (ranging from 0 to 80) [18,19]. Finally, the CT-adapted Leaman score, including information on lesion localization, plaque composition (with a multiplication factor of 1 for calcified plaques and of 1.5 for non-calcified and mixed plaques), and the degree of stenosis (with a multiplication factor of 0.615 for non-obstructive (<50% stenosis) and a multiplication factor of 1 for obstructive (≥50% stenosis) lesions) was used to obtain a comprehensive assessment of plaque burden [20].

### 2.3. Blood Sample Collection

Peripheral venous blood samples were drawn prior to CCTA into Vacutainer tubes (Becton-Dickinson, Franklin Lakes, NJ, USA) for hematological exams and into Tempus Blood RNA tubes containing RNA stabilizing reagents (Applied Biosystems, Bedford, MA, USA) for RNA analyses. Indeed, contrast agents would have strongly affected blood-cell gene expression levels and host metabolic/inflammatory responses. Tempus tubes were vortexed for at least 10 s to stabilize RNA and stored at −80 °C until use.

### 2.4. RNA Isolation and Sequencing

Total RNA was isolated using a KingFisher Flex Purification system and a MagMax for Stabilized Blood Tube RNA Isolation kit (Thermo Fisher Scientific, Waltham, MA, USA) following the manufacturer’s instructions, including DNase treatment for genomic DNA removal. Total RNA concentration and quality were evaluated, respectively, using micro-volume spectrophotometry and microfluidics electrophoresis with an RNA 6000 Nano Assay Kit on a 2100 Bioanalyzer system (Agilent, Santa Clara, CA, USA). α- and β-globin mRNA depletion was performed using the GLOBINclear Human kit (Applied Biosystems). This was followed by poly(A)+ RNA enrichment using the Dynabeads mRNA Direct Micro Kit (Thermo Fisher Scientific) starting from 500 ng of globin-depleted RNA.

Barcoded libraries were prepared using the Total RNA-Seq Kit v2.0 and Ion Express RNA-Seq following the manufacturer’s instructions. Then, 200 bp cDNA fragments were amplified by PCR (16 cycles), with the use of specific Barcode BC primers for library demultiplexing, and quantified on a 2100 Bioanalyzer System. A total of 100 pM diluted libraries were randomly pooled (six samples/pool), loaded on 550 chips, and sequenced with an Ion Gene Studio S5 Prime System (Thermo Fisher Scientific).

### 2.5. RNA-Seq Data Pre-Processing

A mean of 18.5 ± 0.53 million reads per sample was obtained. Sequential aligning of raw reads was performed against the GRCh38 (hg38) human genome. In particular, reads were aligned to the reference genome using STAR [21] and Bowtie 2 [22] aligners. Gene annotation and quantification were computed using FeatureCounts software [23] to obtain a gene expression matrix of raw reads. Raw counts data were then imported into R software v 4.0.0 and filtered to retain genes with a minimum of five counts in at least 40% of the samples.

### 2.6. Differential Expression Analysis

Differential expression analysis was performed with the LIMMA package [24], adjusting the statistical model for “technical batches”. A *p*-value distribution plot was used to test the consistency of differential expression analysis [25]. Genes were deemed as significant at a *p*-value < 0.001. We also computed *p*-values adjusted for multiple testing, using Benjamini and Hochberg’s method to control the false discovery rate (FDR). Heatmaps were drawn with the pHeatmap package using significantly differentially expressed genes.

### 2.7. Gene Set Enrichment Analysis

Gene set enrichment analysis (GSEA) [26] was performed on the entire transcriptome with GSEA software version 4.1.0 (Broad Institute, Cambridge, MA, USA), using Gene Ontology biological processes as reference. We deemed gene sets as significantly associated with a phenotype at an FDR-adjusted *p*-value < 0.05. Then, GSEA results were visualized with an enrichment network of the most significant pathways (FDR < 0.05) using the Enrichment Map Software [27] version 3.3.0, a Cytoscape [28] (version 3.8) plugin.

A similar approach was used to perform cell-type enrichment analysis. To this end, we used a gene-set collection including 22 subsets of human hematopoietic cell types [29].

### 2.8. Microarray Analysis

Transcriptome profiling was repeated in a subgroup of patients using the Clariom D Assay microarrays (Thermo Fisher Scientific), following the manufacturer’s instructions. Gene expression analysis was performed in the R environment v4.0.3, using R/Bioconductor packages. Briefly, probe-level raw data were imported as CEL files, normalized using the robust multichip average (RMA) method [30] implemented in *oligo* [31], annotated with the Affymetrix clariomdhuman annotation data *clariomdhumantranscriptcluster.db*, and summarized to the gene level with *affycoretools*. To control for unwanted heterogeneity, the normalized expression matrix was assessed for the presence of latent variables through the *DaMiRseq* R/Bioconductor package [32]. The statistical model to compare CAD vs. noCAD patients was implemented through the LIMMA package [24], adjusting for a latent variable; i.e., technical batches. To assess the technical reproducibility of cell-type enrichment analysis, Pearson’s correlation was calculated between the mean differences in hematopoietic cell gene sets observed with RNA-Seq and those detected with microarrays.

### 2.9. Statistical Analysis

Continuous variables were presented as means with standard deviation (SD), and categorical data as counts and proportions. Continuous variables normally distributed were compared using Student’s *t*-test for independent samples. The proportion of the categorical variables was compared using a χ^2^ analysis or Fisher’s exact test, as appropriate. Linear regression analysis was performed to evaluate the relationships between each expressed gene and CCTA high-risk features. *p*-value distribution plots were used to test the consistency of the associations observed. A *p*-value < 0.05 was considered statistically significant, except where indicated. Statistical analysis and graphics were produced with native R functions.

## 3. Results

### 3.1. Study Population

For this pilot study, we selected 54 patients, either with severe coronary stenosis (≥70% in at least one vessel; CAD, *n* = 27) or without atherosclerosis (noCAD, *n* = 27), matched for sex, age, and CV risk factors using the function *GetSetMatched* embedded in the R/Bioconductor package CGEN. Two noCAD subjects were excluded from the analysis because of the poor quality of the RNA extracted (RNA Integrity Number [RIN] < 5). Thus, the final, cohort was composed of 52 patients (CAD, *n* = 27; noCAD, *n* = 25).

Table 1 shows the characteristics of the patients under study.

### 3.2. Differential Gene Expression between CAD and NoCAD Patients

Following alignment, read count, and data filtering, 15,128 expressed genes were identified. Genes expressed in peripheral blood were used to evaluate their ability to discriminate between the two phenotypes under study. We observed 2949 differentially expressed genes at a nominal *p*-value < 0.05 with a log_2_ fold-change (FC) range between −0.96 and 0.93. Of these, 1507 (51%) were on average more expressed in CAD subjects, while 1442 (49%) were less expressed in CAD subjects (Appendix A). Applying more stringent criteria, there were 138 significantly differentially expressed genes (*p*-value < 0.001) with a log_2_ FC ranging from −0.63 to 0.75. Of these 138 genes, 74 (54%) showed a higher expression in CAD subjects, while 64 (46%) had a lower expression. The volcano plot in Figure 1 shows the extent of the differential gene expression between CAD and noCAD subjects.

### 3.3. Hierarchical Clustering

Unsupervised hierarchical clustering analysis was performed to evaluate if the genes differentially expressed were able to distinguish CAD from noCAD subjects. The dendrogram (Figure 2) shows a hierarchical organization with two main clusters, highlighting a good separation between CAD and noCAD subjects. In fact, 81% of CAD subjects (*n* = 22) are grouped into a single cluster and 76% (*n* = 19) of noCAD subjects constitute a second well-separated group.

### 3.4. Functional Inference

We performed GSEA using the collection of gene sets reported in the Gene Ontology database of biological processes (GO-BP) to determine whether members of a set of genes tend to be better associated with one phenotype than another (Appendix A). Fifty gene pathways were positively associated with CAD, while 300 pathways were negatively associated (FDR-adjusted *p*-value < 0.05). The most significant GO-BP terms are reported in Table 2.

The results obtained through the GSEA analysis were graphically displayed in an enrichment network, which highlights the most significant pathways and the relationships between them (Figure 3). The coronary atherosclerotic stenotic phenotype was positively associated with tRNA processing pathways, ribosomal metabolic process and assembly, mitochondrial transport, complement activation, and B cell-mediated immunity. Conversely, the stenotic atherosclerosis phenotype was negatively associated with pattern recognition receptor signaling pathway, regulation of innate immunity and inflammatory responses, and angiogenesis.

Through the GSEA, we also carried out a cell enrichment analysis (Appendix A) using a set of hematopoietic cell genes. We sought possible relationships between specific cell types of the immune system and the atherosclerotic phenotype, based on the expression profiles of circulating cellular poly(A+) RNA. The analysis showed a clear distinction between the enriched cell types in the two phenotypic groups (Figure 4). Indeed, the CAD group was found to be enriched by gene expression profiles associated with T lymphocytes (Tregs, T-CD4^+^, T-CD8^+^, and follicular T cells) and B lymphocytes (naive and memory cells). All these cell types play important roles in the adaptive immune system. Conversely, the CAD group was found to be negatively enriched by all cell types associated with the innate immune system, such as granulocytes (both neutrophils and eosinophils), monocytes, and activated and resting mast cells.

To confirm these findings, we selected a subgroup of patients (CAD *n* = 14 vs. noCAD *n* = 13, matched for sex, age, and cardiovascular risk factors) and repeated the differential gene expression analysis using microarrays. Then, we correlated the mean differences (log_2_ FC) in hematopoietic cell gene sets observed with RNA-Seq to those detected with microarrays. The analysis revealed highly significant correlations between the two independent detection methods for each hematopoietic gene set (with Pearson’s correlation coefficients ranging from 0.53 to 0.86 and all *p*-values < 0.01; Appendix A) and strongly supported the observations on phenotype-specific cell enrichment.

### 3.5. Associations between Gene Expression and High-Risk Plaque Features

Linear regression analysis was performed to assess possible associations between gene expression in whole blood and high-risk features detected at CCTA. Genes were considered significantly associated with a *p*-value < 0.01 and |R| ≥ 0.6. The robustness of the regression analysis results was assessed by exploring the histograms of the *p*-value distribution. Ideally, *p*-values should show a uniformly flat distribution across the unit interval (null *p*-values) with a peak near the zero value (*p*-values for alternative hypothesis). Notably, regression analysis showed reliable *p*-value distributions for Leaman score, SIS, SSS, and plaques with density < 150 HU in CCTA (Appendix A). Regression analysis, in contrast, did not detect consistent associations between gene expression and other high-risk features (NRS, LAP, spotty calcification, RI > 1.4) and total plaque volume (Appendix A).

The numbers of genes significantly associated with the Leaman score, SIS, SSS, and plaque density < 150 HU are reported in Table 3. The genes associated with these four CT traits, along with their significance levels, are shown in Appendix A. These may represent robust transcriptional predictors of the four high-risk features in CCTA, both positively and negatively correlated.

## 4. Discussion

The main findings of the present study suggest (*a*) that patients with significant coronary stenosis (≥70%) at first diagnosis are characterized by whole-blood transcriptome profiles that can discriminate them from patients without coronary plaques as detected by CCTA and (*b*) that blood-based transcriptional signatures may predict specific high-risk CT plaque features. These results were derived from genome-wide transcript profiling in the whole blood of a group of 54 patients—27 with CCTA-documented CAD and 25 without signs of atherosclerotic lesions—matched for sex, age, and CV risk factors. Clustering analysis highlighted that the expression patterns identified could distinguish the two phenotypic groups, and functional enrichment analysis showed specific pathways and cell subpopulations positively or negatively associated with the CAD phenotype.

Genome-wide molecular profiling (specifically peripheral blood gene expression profiling) is an informative tool for investigating disease states and identifying markers reflecting genetic predisposition and/or disease activity. Blood is an ideal surrogate tissue for CAD studies [10,11] because it includes immune and inflammatory cells that are key elements in the atherosclerotic process. The search for transcriptional signatures in the whole blood, rather than in cell fractions, reduces handling artifacts (such as induction of transcriptional changes due to cell fractionation procedures) and sample-to-sample variability. Previous studies reported gene expression patterns in peripheral blood that correlate with the extent of CAD [10,33,34]. However, profiling was performed using microarrays, which are limited in dynamic range. Only recently have a few studies used RNA-Seq to look for transcriptional markers of the degree of coronary stenosis or related conditions, although in all of these the vessels were being investigated using invasive angiography in symptomatic patients [12,13,14,15].

RNA sequencing has unveiled new molecular players that could be useful for an in-depth understanding of the pathophysiology of CAD. The molecular and cellular mechanisms underlying coronary atherosclerosis include intracellular lipid accumulation and immune-inflammatory response, which play crucial roles in all the different stages of the atherosclerotic process (i.e., initiation, perpetuation, and resolution of the atherosclerotic process) [35,36]. In addition, increased reactive oxygen species production and mitochondrial dysfunction (with subsequent release of damage-associated molecular patterns and apoptosis-triggering molecules) contribute to atherogenesis, and it has been reported that mitochondrial DNA mutations could play a role in disease progression, although the mechanism remains unknown [37]. Our data showed that the coronary atherosclerotic phenotype was positively associated with mitochondrial respiratory processes, electron transport chain, complement activation, and B-cell-mediated immunity and, in contrast, negatively associated with pattern recognition receptor signaling pathway and regulation of innate immunity and inflammatory responses.

Accordingly, cell-type enrichment analysis revealed that CAD and noCAD groups had a divergent association with the immune response. The CAD phenotype was associated with transcriptional profiles of T lymphocytes (Tregs, T-CD4^+^, T-CD8^+^, and follicular T cells) and B lymphocytes (naive and memory cells), whereas the noCAD group was enriched with cell types associated with the innate immune system, such as granulocytes, monocytes, and mast cells. Although atherosclerosis is considered a chronic inflammatory disorder whose progression is driven by the innate immune response, with the principal involvement of myeloid cells, the hypothesis that both innate and adaptive immunity are involved in the development of this disease has emerged from recent studies [38], and involvement of regulatory T cell (Tregs) imbalance has been described [14]. The presence of B cells and T cells at the level of the lesion suggests that atherosclerotic pathology may also be characterized by the presence of an important autoimmune component [28,38].

Hence, cellular enrichment of the different types of T and B lymphocytes in this study could suggest an association between a systemic antigen-specific immune component and the stenotic atherosclerotic phenotype, supporting the hypothesis of an auto-immune response in CAD subjects with advanced pathology. Our data suggest that, in a late phase of the atherosclerotic process (overt stenotic CAD), the autoimmune response is prevalent over the innate inflammatory processes. Consistent with this hypothesis, in a recent study [39] that investigated whole-blood transcriptome profiles in patients with acute myocardial infarction, we observed that the non-ST-segment elevation acute myocardial infarction (NSTEMI) phenotype was associated with immune cells, such as T and NK cells, and with leukocyte/lymphocyte activation processes.

Finally, this study suggests the existence of peripheral blood gene expression patterns able to predict the degree of some high-risk features in CCTA. Of interest, regression analysis unveiled several genes whose expression was consistently correlated with the Leaman score, SIS, and SSS, as well as plaques with density <150 HU in CCTA. Indeed, the combination of these parameters allows a comprehensive assessment of atherosclerotic burden, as SIS represents the diffuseness of CAD, SSS combines the extent of disease with the degree of stenosis, the Leaman score points out plaque characteristics and lesion proximity, and plaque volume < 150 HU is focused on plaque composition. In contrast, no association was found with the total coronary plaque volume, LAP, NRS, spotty calcification, and RI > 1.4. LAP (expression of the lipidic composition of the plaque), NRS (representing a plaque with a large necrotic core), spotty calcification (associated with intraplaque hemorrhage), and RI > 1.4 (expression of early atherosclerosis) are qualitative plaque features previously associated with increased risk of ACS [2]. However, our findings are in line with the major results of recent prospective, multicenter studies on the prognostic value of plaque characterization in CCTA, such as CAPIRE and ICONIC, that demonstrated the superiority of CCTA-derived quantitative vs. qualitative parameters in predicting acute coronary events [40,41]. Both these studies showed that quantitative plaque parameters, such as noncalcific (<150 HU), otherwise called fibro-fatty, plaque volume, are superior in stratifying patient prognosis and predicting ACS than the presence of qualitative high-risk plaque features, such as LAP, NRS, spotty calcification, and positive remodeling. On the other hand, more recent insights from CAPIRE showed a strict association between the noncalcific component (<150 HU) of the plaque volume (the highest quartile of the noncalcific plaque volume) and inflammatory biomarkers such as high-sensitivity CRP or pentraxin 3, whereas no significant association was found between inflammatory biomarkers and elevated total coronary plaque volume [16]. These data support the role of inflammation in atherosclerosis pathophysiology, but the absence of an association between inflammatory biomarkers and total coronary plaque volume may suggest that inflammation selectively enhances high-risk atherosclerosis development, but it is not directly involved in the global coronary atherosclerosis burden [16]. Notably, CCTA, particularly when the latest scanner generation equipped with improved spatial and temporal resolution is used, has been demonstrated to accurately measure coronary plaque volume in comparison with the invasive standard of reference [42]. Indeed, a recent study showed a mean difference between plaque volume defined by IVUS and CCTA performed with a 256-slice scanner of 4 mm^3^ only, with boundaries of agreement of 21.8 and −13.9 mm^3^ and a very high correlation coefficient (0.98) [17].

Thus, an association between transcriptional profiles in the whole blood and plaque characteristics could help early identification of the “vulnerable patient” and, therefore, prevent CV events, as previously suggested in the literature [10].

The strength of these results is that they were obtained from whole blood and with an RNA sequencing method, increasing the reliability and robustness of emerging biomarkers for the management and stratification of patients. The small size of the population represents a limitation of this study and it will be necessary to confirm the results on a larger number of patients and validate them with a new independent sample group. In addition, it should be considered that the sequencing analysis of the circulating transcriptome was carried out on all cell populations present in the peripheral blood (bulk analysis) and, therefore, it was not possible to associate a specific pathway with a precise cellular subpopulation.

## 5. Conclusions

The results of this explorative pilot study show differences in the expression of the circulating transcriptome between patients with significant coronary stenosis (≥70%) and subjects with the absence of coronary atherosclerosis in CCTA. These different patterns are functionally associable with two divergent immune-inflammatory profiles. Moreover, the results of regression analysis suggest the hypothesis that a possible association between whole-blood transcriptional profiles and plaque characteristics could help identify the “vulnerable patient” and potentially prevent CV events.

## Figures and Tables

**Figure 1 biomedicines-10-01309-f001:**
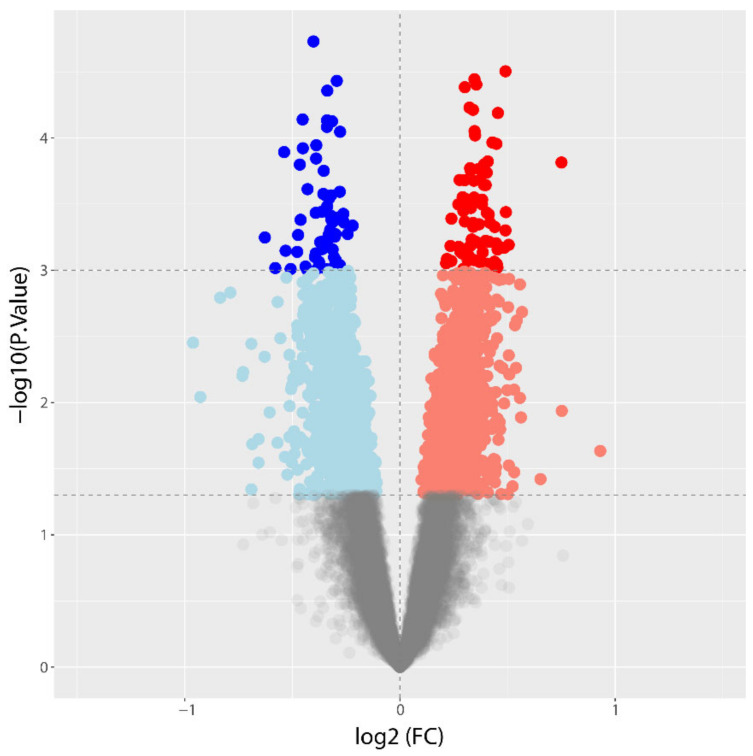
Volcano plot of differential gene expression analysis of CAD vs. noCAD patients. Red and blue dots represent genes significantly overexpressed or decreased in CAD patients, respectively (*p* < 0.001). Pink and light blue dots highlight genes with higher or lower expression in CAD vs. noCAD patients, respectively, at a nominal *p*-value < 0.05. Grey dots represent genes with no difference in expression between the two groups.

**Figure 2 biomedicines-10-01309-f002:**
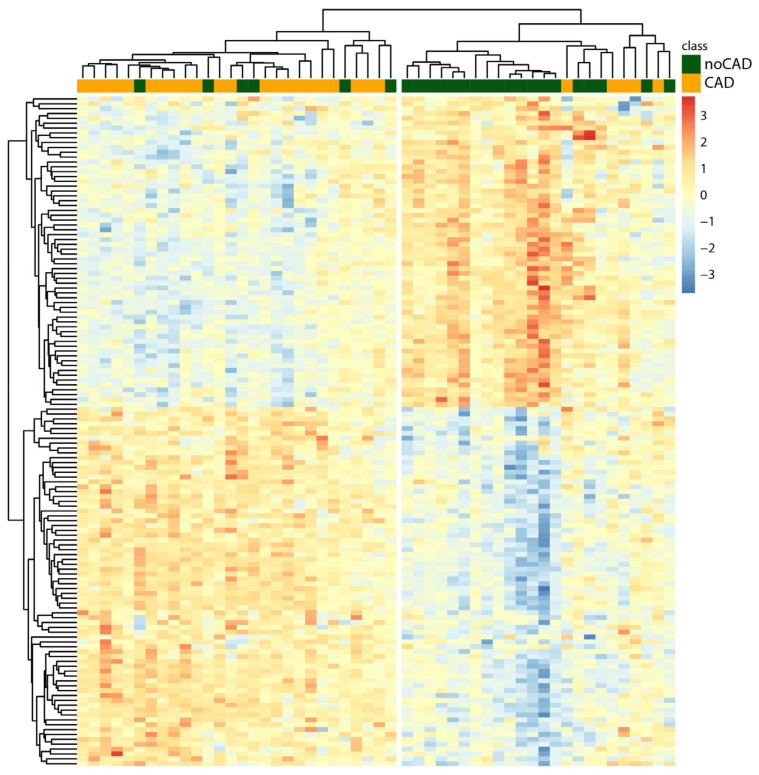
Heatmap visualizing hierarchical clustering results. The dendrograms show the subjects under study in the columns and the differentially expressed genes in the lines. In the heatmap, each gene is associated with a chromatic index indicating normalized expression in the sample (from bright blue = low level of expression to dark red = high level of expression).

**Figure 3 biomedicines-10-01309-f003:**
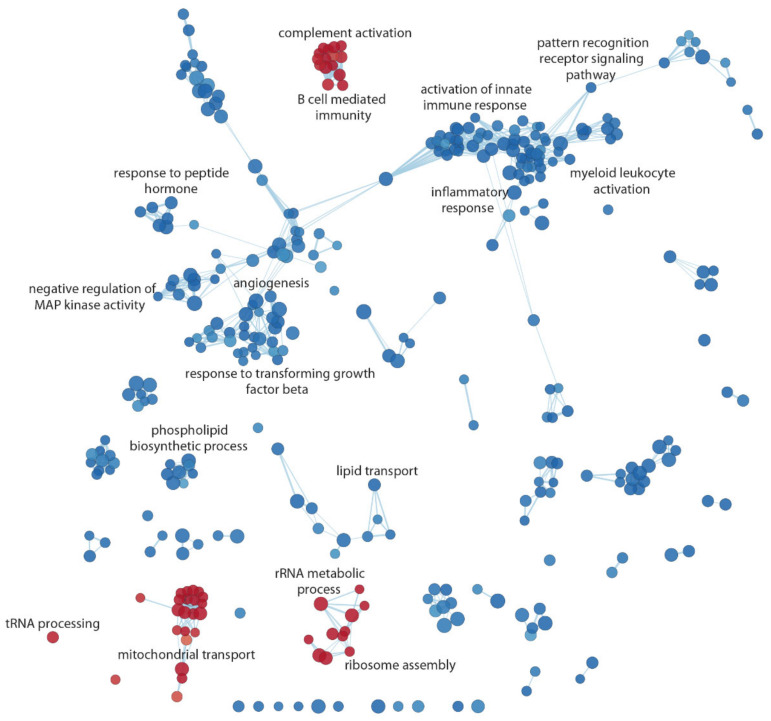
Enrichment map of the gene sets positively (red nods) and negatively (blue nodes) associated with CAD, respectively. The network shows the most significant results of the GSEA with the Gene Ontology biological processes gene sets (FDR-adjusted *p*-value < 0.05).

**Figure 4 biomedicines-10-01309-f004:**
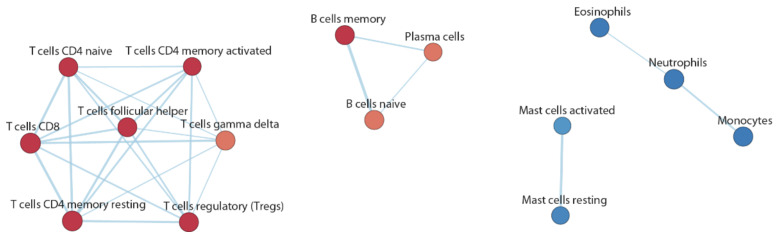
Cell enrichment map. The red nodes represent the enriched cells in the CAD phenotype, while the blue nodes were enriched in the noCAD phenotype and negatively associated with CAD.

**Table 1 biomedicines-10-01309-t001:** Patient characteristics.

Clinical Characteristics	CAD (*n* = 27)	noCAD (*n* = 25)	*p*-Value
Males	21 (77.8%)	20 (80%)	0.84
Age (years)	63 ± 8	62 ± 7	0.65
BMI (kg/m^3^)	26.2 ± 3	26 ± 2.7	0.8
Smokers	16 (59.2%)	13 (52%)	0.60
Hypertension	19 (70%)	10 (40%)	0.045
Hypercholesterolemia	19 (70%)	16 (64%)	0.64
Use of statin	15 (55%)	7 (28%)	0.06
Diabetes mellitus	4 (14.8%)	3 (12%)	0.76
Peripheral artery disease	5 (19%)	2 (8%)	0.42
** *Clinical presentation* **			
Any of the symptoms below	15 (55.5%)	13 (52%)	0.80
Angina pectoris	3 (11%)	2 (8%)	0.71
Atypical chest pain	6 (22%)	4 (16%)	0.58
Dyspnea	3 (11%)	2 (8%)	0.71
Arrhythmias	4 (14%)	7 (28%)	0.21
** *Laboratory* **			
Erythrocytes (10^6^/µL)	4.88 ± 0.5	4.91 ± 0.41	0.81
Leucocytes (10^3^/µL)	8.32 ± 1.67	7.67 ± 1.84	0.19
Hemoglobin (g/dL)	14.83 ± 1.42	14.85 ± 1.12	0.95
Hematocrit (%)	43.04 ± 3.91	43.50 ± 2.67	0.62
Platelets (10^3^/µL)	239.88 ± 50.38	255.48 ± 55.82	0.33
Glycemia (mg/dL)	100.33 ± 10.9	102 ± 27.27	0.77
Uric acid (mg/dL)	5.34 ± 1.05	5.2 ± 1.40	0.68
γ-GT (UI/L)	35.34 ± 22.35	34.08 ± 23.9	0.84
Total bilirubin (mg/dL)	0.62 ± 0.27	0.71 ± 0.34	0.31
Troponin I (ng/L)	5.07 ± 11.2	2.81 ± 2.23	0.31
Triglycerides (mg/dL)	109 ± 65.82	94.44 ± 33.64	0.31
Total cholesterol (mg/dL)	200.48 ± 49.26	195.84 ± 40.42	0.71
HLD-c (mg/dL)	58.22 ± 13.37	66.76 ± 18.05	0.06
LDL-c (mg/dL)	120.40 ± 41.59	110.08 ± 33.99	0.33
CRP (mg/dL)	2.12 ± 2.59	1.46 ± 1.90	0.30

Categorical variables are presented as counts (*n*) and proportions (%); quantitative variables are expressed as means ± SD. Continuous variables were compared using Student’s *t*-test for independent samples. The proportions of the categorical variables were compared using Fisher’s exact test. SD: standard deviation; BMI: body mass index; γ-GT: γ-glutamyl transpeptidase; HDC-c: high-density lipoprotein cholesterol; LDL-c: low-density lipoprotein cholesterol.

**Table 2 biomedicines-10-01309-t002:** Top non-redundant Gene Ontology biological processes associated with CAD.

NAME	Gene Ontology ID	NES	q-Value
** *Positively associated* **			
rRNA metabolic process	GO:0016072	4.706	0
Aerobic respiration	GO:0009060	3.535	0
Ribosome assembly	GO:0042255	3.732	0
Complement activation	GO:0006956	2.956	0.00007
B cell-mediated immunity	GO:0019724	2.686	0.00053
Mitochondrial transport	GO:0006839	2.490	0.00185
tRNA processing	GO:0008033	2.070	0.02193
** *Negatively associated* **			
Pattern recognition receptor signaling pathway	GO:0002221	−3.129	0
Negative regulation of MAP kinase activity	GO:0043407	−2.894	0.00058
Response to peptide hormone	GO:0043434	−2.719	0.00206
Myeloid leukocyte activation	GO:0002274	−2.644	0.00243
Inflammatory response	GO:0006954	−2.544	0.00399
Activation of innate immune response	GO:0002218	−2.529	0.00422
Response to transforming growth factor-beta	GO:0071559	−2.384	0.00694
Angiogenesis	GO:0001525	−2.316	0.00956
Phospholipid biosynthetic process	GO:0008654	−2.197	0.01545
Lipid transport	GO:0006869	−2.032	0.02875

NES: normalized enrichment score; q-value: false discovery rate-adjusted *p*-value.

**Table 3 biomedicines-10-01309-t003:** Numbers of genes significantly associated with CT plaque features (*p* < 0.01 and |R| ≥ 0.6).

Variable	Significantly Associated Genes	Positively Associated Genes	Negatively Associated Genes
Leaman Score	19	11	8
SIS	34	17	17
SSS	17	7	10
Plaque density < 150 HU	58	42	16

## Data Availability

The RNA-Seq raw and normalized data used in the current study are publicly available in the NCBI GEO repository under the accession number GSE202625 (https://www.ncbi.nlm.nih.gov/geo/query/acc.cgi?acc=GSE202625). Other data presented in this study are available in Appendix A.

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
