# Peer review of "Whole-Blood Transcriptional Profiles Enable Early Prediction of the Presence of Coronary Atherosclerosis and High-Risk Plaque Features at Coronary CT Angiography"

_biomedicines, 2022, doi:10.3390/biomedicines10061309_

Round 1
Reviewer 1 Report
This article is devoted to the study of blood transcriptome in patients with coronary atherosclerosis with the goal to find biomarkers for disease detection.
Overall the paper is well written, the results obtained by the authors support the conclusions.
Considering above mentioned, we should also mention certain issues.
List of abbreviations used would benefit the manuscript.
Introduction. The authors should add some additional information on atherosclerosis pathogenicity - to demonstrate that there are different types (or stages) of atherosclerotic lesions. It would be beneficial to mention hypothesis related to atherosclerosis development, for example, the roles of multiple modified LDL, including oxidised LDL, and often forgotten desialylated LDL, deserve to be discussed, as well as the intrigue related to the potential involvement of mutations of mitochondrial DNA in pathogenicity of atherosclerosis. The addition of a short information on roots or causes of atherosclerosis development would strengthen the Introduction. Some insights related to above mentioned issues could be found, for example, in several publications mentioned here (or in another publications on the subject which will be found by the authors in PubMed): DOI: 10.3389/fcvm.2021.707529 ; DOI: 10.3390/biomedicines9060600 ; DOI: 10.3390/ijms22084080. It could help later with Discussion section, especially in relation to the found association of CAD with elevated expression of some mitochondrial genes.
Line 191: typo - Table 2 instead of Table 1. Type of statistical tests applied should be mentioned in Table 1 legend.
Section 3 (Results). The numbers in subsections should be corrected since number “3.1” repeats in all subsections.
Supplementary Table S1. The column “adj.P.Val” should be explained in the table legend as well as the statistical test applied. What importance do these values (adj.P.Val) have for explanation of results obtained?
Line 231. Supplementary Table S2 was absent in files attached.
Line 251. Supplementary Table S3 also was absent in files attached.
Line 281. There is no Table 5 mentioned in the text. It is probably Table 3. It would be nice to see the exact results (genes) mentioned in Table 5 (3) in Supplementary Table.
Discussion section. The authors could add a little bit more discussion related to their finding about some elevated mitochondrial genes in CAD patient considering existence of association of certain mutations of mtDNA with atherosclerotic lesions.
Line 378. Patents. There is no information there.
The paper can be accepted after minor revision related to the answers to above mentioned questions/suggestions.
Reviewer 2 Report
Authors described whole-blood transcriptomal profiles present in coronary atherosclerosis to predict early stage of the disease, also they explored the features of plaque by CT angiograpy to identified high risk plaques. The overall of the manuscript is well written and the content is of interest for the readerships. However, there are several issues that have to be seriously improved.
Mayor comments
Introduction:
Authors should provied background about blood RNAseq in coronary atherosclerosis with references. There are a few studies published in this context (i.e.MacCaffrey et al., BMC Medical Genomics 2020; Chen et al., ACS Omega, 2021; Zhang et al, BMC Med Gen. 2021, etc.).
Mat&Meth: Authors should indicate the aprox sequecing depth per sample acquired with the sequencing method used.
On "gene set enrichment analysis" authors indicated that they deemed gene sets ... associated with FDR adjusted p-value <0.05, however there are not genes meeting this criteria on their list in supp file table S1. This should be clarified.
Table 1. What "symptoms" means? There are 13 patients of noCAD category indicated as symptoms. It is not clear what that means.
Did the noCAD patients have been investigated for any other type of atherosclerosis (i.e. carotid atherosclerosis)?
Authors report on page 9 that the two phenotypes can be differenciated by the enriched cell types. It would be of high interest to test these findings by, for instance, flow cytometry in a few patients to see that these results are confirmed. Or alternatively, authors could choose some of the key genes associated with those cell types and hve them tested by qPCR.
Part 3.1: The association between gene expression and high-risk plaque features should be better described in the text. What the variables Leamen Score, SIS...in relation with high-risk plaque features, so to be able to understand the results of the shown table. Also, it would be of high interest to know the identity of those genes, maybe on a table as supplementary file?
Minor comments:
Line 140, DNAse should be written as DNase.
Line 189, clarify "poor quality of RNA extracted" (i.e. low RIN number...)
Line 203. What exactly contains table S1?. Authors should indicate it also in the legend. What "t" means on that table? Also they should indicate whta adj P value is and how it was calculated on math&meth.
Line 235. A q-value is a p-value that has been adjusted for false discovery rate (FDR). For that reason it is more correct to call it just "q-value" or "FDR adjusted p-value"
Check the numbers of tables, where are table 3 and 4?
